# Regardless of the Brewing Conditions, Various Types of Tea are a Source of Acetylcholinesterase Inhibitors

**DOI:** 10.3390/nu12030709

**Published:** 2020-03-06

**Authors:** Ewa Baranowska-Wójcik, Dominik Szwajgier, Anna Winiarska-Mieczan

**Affiliations:** 1Department of Biotechnology, Microbiology and Human Nutrition, University of Life Sciences in Lublin, 8 Skromna Street, 20-704 Lublin, Poland; 2Department of Bromatology and Food Physiology, University of Life Sciences in Lublin, Akademicka 13, 20-950 Lublin, Poland; anna.mieczan@up.lublin.pl

**Keywords:** tea, Alzheimer’s disease, AChE

## Abstract

Alzheimer’s disease (AD) is a progressive neurodegenerative disease characterized, among others, by abnormally low levels of the neurotransmitter acetylcholine in the brain. Acetylcholinesterase (AChE) plays a significant role in the process through hydrolysis of the acetylcholine neurotransmitter. Currently, the main method for treatment of AD at a symptomatic stage entails administration of AChE inhibitors to patients diagnosed with the disease. However, it is also possible to take certain steps to treat AD by delivering inhibitors with food. There is a growing body of evidence to suggest that tea (*Camellia sinensis*) shows numerous beneficial properties, including improving cognitive abilities. This is particularly important in the case of AD patients. The study assessed the impact of brewing conditions on the inhibition of AChE activity observed in tea extracts (black, white, or fruit). Our study revealed that neither temperature nor time of brewing influenced the respective infusions’ ability to inhibit the activity of AChE. Anticholinesterase activity was observed in most of the different types of tea that were analyzed, with the highest rate of inhibition (30.46%–48.54%) evidenced in the Biofix Tea Wild Strawberry brand. The results of our research show that tea may be used as a rich source of cholinesterase inhibitors which play a significant role in AD treatment.

## 1. Introduction

Dementia is currently one of the fastest spreading diseases with an estimated 36 million people suffering worldwide. The most common form of dementia is Alzheimer’s disease (AD) [1]. Studies indicate that by 2050, the number of affected patients may reach 115 million, which constitutes a nearly threefold increase relative to the disease’s prevalence in the present decade [2]. 

The pathophysiology of AD is related to the depletion of the acetylcholine (ACh) neurotransmitter [3,4]. Acetylcholinesterase (AChE) and butylcholinesterase (BChE) hydrolyze ACh in the synaptic cleft [5,6], and one of the possible therapeutic strategies for symptomatic treatment entails the use of AChE inhibitors [7,8]. According to the cholinergic hypothesis, AChE inhibitors lead to an increase in the ACh concentration in the brain and improvement of AD patients’ cognitive functions [9,10]. The currently available medicines (rivastigmine, galanthamine, donepezil) are administered in the late stages of the disease [11]. However, the drugs are not without significant limitations. The improvement of cholinergic transmission lasts no longer than three years, whereas inhibitors are effective in symptomatic treatment of AD but have little to no effect on the characteristics modifying the course of the disease [12].

Due to antioxidant activity, it is currently believed that drinks rich in polyphenols are recommendable as ingredients of a natural supplementary therapy, helping to alleviate the symptoms of AD [13]. One of those drinks is tea, including fruit tea variants which have been gaining considerable popularity in recent years due to their attractive aroma, taste, and the health benefits of constituting a natural source of antioxidants [14,15,16]. The ingredients of such tea—fruit, flowers, leaves, and other plant material—are an important source of phenolic compounds, as well as vitamins and minerals. Fruit such as red raspberry (*Rubus idaeus*), cranberry (*Oxycoccus macrocarpos* and *O. palustris*) as well as various wild rose species (*Rosa canina*) are ingredients commonly included in fruit tea to enhance both its taste and health benefits [16]. 

Research suggests that drinking tea on a regular basis can improve speech quality and fluency assessment [17], and can limit the deterioration of cognitive functions [18]. Tea can also alleviate the symptoms of many other diseases and prevent the development of obesity, circulatory diseases, type 2 diabetes, and certain types of cancer (ovary, lung, skin, breast, endometrium, prostate, bladder, oral cavity, and large intestine) [19,20,21,22]. Its potential health benefits have been attributed to the content of phenolic compounds with unique biological properties which constitute 30% of tea dry mass [23]. Polyphenols are the primary active compounds present in tea [24,25,26] which, due to their ability to sweep free radicals, are considered to be strong antioxidants [27,28,29]. Epigallocatechin gallate (EGCG), epigallocatechin (EGC), epicatechin (EC), and rutin are among polyphenols found in tea that show a neuroprotective effect on health and cognitive function. They function by inhibiting the amyloid precursor protein (APP) splitting, preventing incorrect protein folding and membrane damage induced by Aβ. They also suppress Aβ oligomer aggregation, alleviate Aβ-induced oxidative stress, regulate the signal pathway including Aβ aggregation, inhibit microtubule associated protein (TAU) hyperphosphorylation, and inhibit AChE [14,30,31]. The caffeine contained in tea has also been shown to have protective properties against neurological diseases such as AD due to its stimulatory effects on the central nervous system [32].

To date, no reports have been published pertaining to the impact of fruit tea brewing conditions (temperature and time) on the capacity to inhibit AChE activity, hence the aim of the present study was to determine whether said conditions can have a significant influence on anti-AChE activity. The choice of the respective samples was also targeted due to the tea brands’ considerable market share in Poland and their taste quality.

## 2. Materials and Methods 

### 2.1. Chemicals

Acetylcholinesterase (AChE, C3389), acetylthiocholine iodide (ATCh, 01480), 5,5′-dithiobis(2-nitrobenzoic acid) (DTNB, D8130), and Tris-HCl buffer (1185-53-1) were purchased from Sigma-Aldrich (Saint Louis, MO, USA/Poznań, Poland). Folin–Ciocalteu reagent (F-C, 694350111) and other analytical grade reagents were purchased from Avantor (formerly P.O.Ch. Gliwice, Poland).

### 2.2. Materials

In all, 14 different instant tea variants were purchased in 2019 from shops in Lublin, Poland. All data regarding the products, in particular their names and contents, were obtained from product labels (Table 1). Tea infusions were prepared as follows: teabags (2.5 g) were placed in 200 mL beakers filled with fresh water heated up to, respectively, 100, 90, 95, 85, 80, and 75 °C. Next, the beakers were left for 5, 10, or 15 min, respectively, to obtain infusions. Afterwards, the bags were squeezed out and removed. Samples were collected at each brewing time point and stored at −80 °C until the analysis. Each sample was prepared and analyzed in triplicate.

The dry weight of infusions was measured after drying precisely weighed samples of the respective samples (50 ± 0.1 mg) at 105 ± 5 °C, until a fixed mass (maximally 28 h) was reached. 

### 2.3. Determination of ChE Inhibitory Activity

AChE inhibition was tested using a 96-well microplate reader (Tecan Sunrise, Grödig, Austria) based on the method developed by Ellman et al. [33], with certain modifications [34]. The test solution was composed of 0.035 cm^3^ of the studied sample, 0.035 cm^3^ ATCh (1.5 mM/mL), 0.175 mL of 0.3 mM/mL 5,5’-dithiobis-(2-nitrobenzoic acid) (DTNB, containing 10 mM/mL NaCl and 2 mM/mL MgCl_2_), and 0.02 mL of the AChE or BChE solution (0.2 U/mL). The volume of the sample was filled up to 0.345 mL using Tris-HCl buffer (100 mM/mL, pH 8.0). All reagent solutions used in this test were prepared in the same buffer. Samples containing Tris-HCl buffer instead of the studied samples were run in the same way (negative samples). The absorbance (405 nm, 22 °C) was read after 30 min. The increase in absorbance due to spontaneous hydrolysis of the substrate was monitored using blank samples containing DTNB and ATCh filled up to 0.345 mL with Tris-HCl buffer. The absorbance originating from the blank sample was subtracted from the absorbance of the test sample. The false-positive effect of samples was tested according to Rhee et al. [35], with minor modifications, as described previously [34]. After mixing the substrate with the enzyme and buffer, the false-positive sample was left for incubation. Then, the studied sample and DTNB were added, followed by an immediate measurement of absorbance. Each sample was analyzed in at least eight replicates and all solutions used in a set of analyses were prepared in the same buffer. The obtained data were expressed as mean (± standard error of mean (SEM)). 

### 2.4. Statistical Analysis

The routine statistical tests including average values and standard deviation were tested. Statistical differences were calculated using Tukey’s HSD (Tukey’s honest significant difference) test with significant differences identified at *p* < 0.05. All tests were performed using Statistica Software (Version 13.1 StatSoft, Cracow, Poland).

## 3. Results

Our study revealed that the brewing conditions did not significantly affect the infusion’s anti-cholinesterase activity. We observed that brewing at higher temperatures 85-100 °C resulted in more effective inhibition of anti-cholinesterase activity regardless of the brewing time, but the differences were not significant compared to results obtained for lower brewing temperatures (Figure 1, Appendix A). The highest level of anti-AChE activity (30.46%–48.54%) was observed for Biofix Tea Wild Strawberry infusions (no. 3) at all brewing temperatures. The lowest activity levels (0.46%–5.09%) were observed for Biofix Tea Cranberry infusion (no. 4). Low activity was also registered for the types of tea numbered 1, 2, and 5. The variation in terms of AChE activity in other brands resulted from differences in terms of tea types and fruit additives. 

## 4. Discussion

Studies conducted worldwide confirm the existence of a correlation between drinking tea and positive effects on human cognitive function [14,17,36]. In their literature review, Polito et al. [14] analyzed epidemiological data and concluded that drinking tea can reduce the risk of AD and other neurodegenerative diseases, as well as improve cognitive functions in the elderly. Okello et al. [37] demonstrated that green and black tea extracts can improve the efficiency of the cholinergic system, which may substantially contribute to alleviating the effects of the cholinergic deficit observed in AD and other age-related memory disorders. In the past, authors demonstrated that both green and black tea are capable of inhibiting the activity of human AChE at IC50 levels of 0.03 mg/mL and 0.06 mg/mL, respectively [37]. Fei et al. [38] observed anti-aging and antioxidative properties of pure tea, namely black and green tea extracts. They noticed that aqueous extracts of the analyzed tea were capable of extending the lifetime of the AD transgenic worm and increased its tolerance to oxidative stress induced by Cr+6 ions. Raghavendra et al. [39] reported that green tea extracts can show strong anti-AChE activity *in vitro*. Kwak et al. [40] studied the impact of various storage conditions on AChE inhibition for green tea. They demonstrated that storage at various temperatures (room temperature, 4 and −20 ℃) did not influence AChE inhibition. In the course of an almost six-year study conducted on a group of patients who were 65 years old or older, Tomata et al. [20] observed that drinking green tea was significantly correlated with a lowered risk of dementia. In a study on mice, Chan et al. [41] observed that the administration of green tea (1% in diet) resulted in an improvement in terms of cognitive functions compared to the control group. 

The beneficial effects of tea most likely stem from the wide range of bioactive compounds present therein [42]. The authors associated the strong anti-AChE activity with a high polyphenol content [43]. Green tea polyphenols (GTPs) induce a wide range of biochemical and pharmacological effects preventing diseases related to oxidative stress [44]. Chung et al. [44] concluded that polyphenols contained in green tea are strong AChE inhibitors with an IC50 of 248 μg/mL. To the authors’ knowledge, the publication was the first report demonstrating GTP’s ability to inhibit AChE activity. Wobst et al. [45] demonstrated that polyphenols such as EGCG present in green tea inhibited aggregation of a TAU protein fragment (K18DK280) in vitro. Bastianetto et al. [46] demonstrated in their research that green and black tea extracts protected cultured hippocampus cells against toxicity induced by amyloid-β (Aβ). The researchers suggest that EGCG and its derivatives may facilitate the treatment of neurological disorders by inhibiting the formation of Aβ fibrils. Schimidta et al. [47] studied the impact of green, red, and black tea supplementation in a rat model of AD. They concluded that green tea was more effective than the other two types, which they attributed to its higher EGCG content. The neuroprotective effects of green tea were also reported by Flores et al. [48]. After its administration to rats, it significantly reduced the levels of reactive oxygen species and induced antioxidant protection. Takahashi et al. [49] observed that a fraction of aromatic green tea displayed antioxidative properties, which may constitute another preventive factor against AD. Pérez-Burillo et al. [42] demonstrated that brewing white tea leaves at 98° C for 7 min yielded the highest content of bioactive compounds in the infusion and the strongest antioxidant activity.

To our knowledge, our report is the first to present a broad comparison of anti-AChE activity of a considerable number of types of fruit tea. In particular, no such comparison was reported concerning types of fruit tea present on the Polish market. Our findings revealed their high potential in terms of lowering the activity of the analyzed enzyme, which encouraged us to continue research on such types of tea. It is highly plausible that, similarly to black and white tea, a key factor in this context is the content of phenolic compounds, which in the case of fruit tea is strongly dependent on the types of fruit used and share thereof in the overall tea mixture [26]. Pękala et al. [50] demonstrated that certain fruit tea types contained significant amounts of flavonoids (naringin and hesperidin) which were not present in premium black tea. In another study, Pękal et al. [25] reported that fruit tea contained higher amounts of chlorogenic and caffeic acid than black tea. Furthermore, citrus and fruit tea proved to be a rich source of naringin and hesperidin.

## 5. Conclusions

Nutrition and correct diet influence overall health and impact the development of central nervous system disorders. Dietary patterns may prove significant to preventing the deterioration of cognitive functions and onset of dementia in later life. The capacity to combine pharmaceutical and non-pharmaceutical means is an important factor in the efforts to slow down the development of Alzheimer’s disease. Hence, it is very important to employ dietary supplementation patterns that facilitate the reduction and alleviation of the disease’s symptoms. Given the role of AChE and the need to inhibit the enzyme’s activity in effective AD treatment, the results presented above may prove valuable and supplementation using tea, including its fruit variants, may prove a viable strategy in AD prophylactics. The effects of bioactive tea ingredients such as flavonoids, catechins, or phenolic acids promises a considerable potential in the context of improving cognitive function or reducing the extent of cognitive disorders in AD patients.

The results of our study encourage us to continue working in this very important research problem in the future, with a particular emphasis on gaining insights into the ability of fruit tea products to inhibit AChE.

## Figures and Tables

**Figure 1 nutrients-12-00709-f001:**
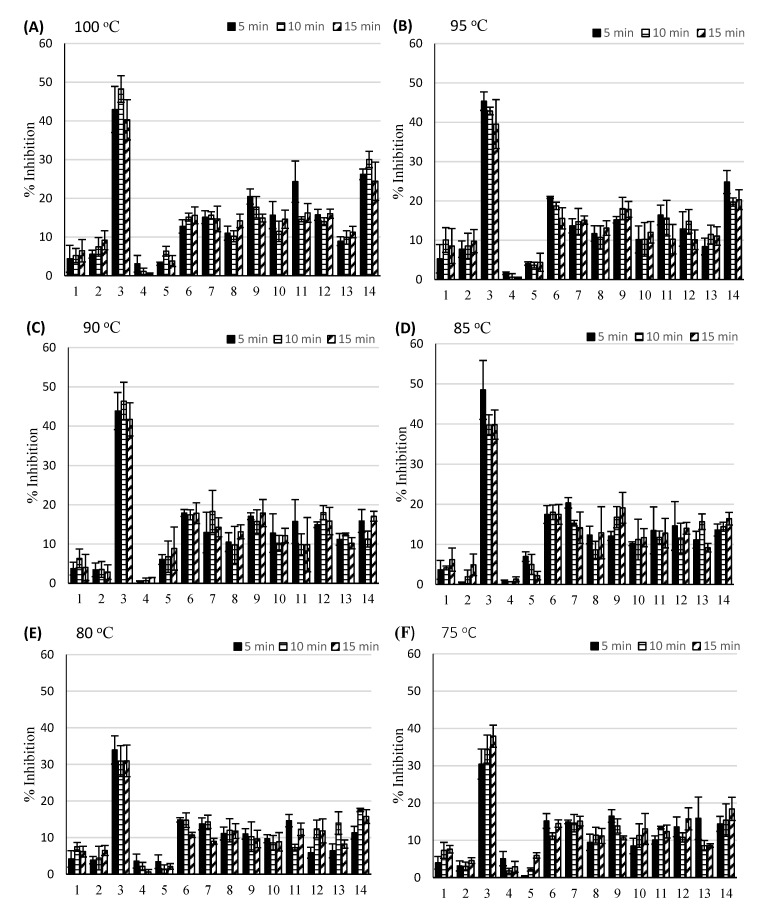
Acetylcholinesterase (AChE) inhibition (%) in the studied teas (averaged results ± standard deviation). (**A**–**F**) represent brewing temperatures. Legends: 1- Biofix Tea Raspberry, 2- Biofix Tea Multi-fruit, 3- Biofix Tea Wild Strawberry, 4- Biofix Tea Cranberry, 5- Biofix Tea Wild Rose, 6- Biofix Green Tea with Quince, 7- Biofix Green Tea Guarana and Passion, 8- Biofix Green Tea Ginseng and Pomegranate, 9- Biofix Green Tea Original, 10- Teekanne White Tea, 11- Irving White Tea Pomegranate and Gooseberries, 12- Tetley Black Tea, 13- Lipton Black Tea, 14- Saga Black Tea.

**Table 1 nutrients-12-00709-t001:** Analyzed tea samples.

No.	Name, Type of Tea (in Bags)	Ingredients
1	Biofix Tea Raspberry	Raspberry fruit, hibiscus flower, chokeberry fruit, apple fruit, chokeberry juice concentrate, aroma, citric acid.
2	Biofix Tea Multi-fruit	Chokeberry fruit, apple fruit, hibiscus flower, raspberry fruit, blackcurrant fruit, orange peel, chokeberry juice concentrate, aroma, citric acid.
3	Biofix Tea Wild Strawberry	Hibiscus flower, wild strawberry fruit, chokeberry fruit, blackcurrant fruit, apple fruit, chokeberry juice concentrate, aroma, citric acid.
4	Biofix Tea Cranberry	Cranberry fruit, hibiscus flower, chokeberry fruit, apple fruit, chokeberry juice concentrate, aroma, citric acid.
5	Biofix Tea Wild Rose	Rosehip fruit, hibiscus flower, apple fruit, fruit chokeberry, chokeberry juice concentrate, aroma, citric acid.
6	Biofix Green Tea with Quince	Green tea, aroma, quince fruit.
7	Biofix Green Tea Guarana and Passion	Green tea, aromas, guarana, passion fruit.
8	Biofix Green Tea Ginseng and Pomegranate	Green tea, aromas, ginseng root, lemon grass, orange peel, pomegranate peel.
9	Biofix Green Tea Original	Green tea original.
10	Teekanne White Tea	White tea.
11	Irving White Tea Pomegranate and Gooseberries	White tea, apple, gooseberry flavor, blackberry leaf, rosehip, pomegranate peel, chicory root, lemon peel, hibiscus flower, acidity regulator, citric acid, liquorice root, pineapple passion fruit, papaya juice concentrates.
12	Tetley Black Tea	Black tea.
13	Lipton Black Tea	Black tea.
14	Saga Black Tea	Black tea.

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
