# Peer review of "Regardless of the Brewing Conditions, Various Types of Tea are a Source of Acetylcholinesterase Inhibitors"

_nutrients, 2020, doi:10.3390/nu12030709_

Round 1

Reviewer 1 Report

General Comments:

The paper reports on the acetylcholine esterase (AChE) inhibitory potential of various types of tea, which was found to be independent from brewing temperature and time. Considering that, in general, the types and quantities of molecules extracted from plant material strongly depend on the extraction conditions, whereas, on the other hand, the brewing of tea is usually done in a non-standardized way, these findings seem important with regard to making use of the health effects of tea.

It is important to strictly distinguish throughout the paper between prevention of Alzheimer's disease (AD) and the symptomatic treatment of AD. Polyphenols and a few other types of molecules, mainly due to anti-oxidant activity or interference with the amyloid cascade, are supposed to have some preventive or disease-modifying effects. AChE inhibition does not seem to have a role in AD prevention or disease-modification, because the known AChE inhibitors did not delay or prevent dementia in controlled clinical trials. During both the introduction and the discussion, the wording should be clear for every substance or substance group, whether it is supposed to have a role in prevention or in symptomatic treatment of dementia. Currently, AChE inhibition has its role in symptomatic treatment of the dementia stage of AD.

Taking into account that the focus of the experimental part of the paper is on AChE inhibition by various types of tea, both the introduction and the discussion should focus more on AChE inhibition than on other mechanisms. Nevertheless, for putting the findings into context, the other beneficial effects of tea should also be mentioned.

Specific Comments:

  1. Abstract, line 15: It is not quite correct that administration of AChE inhibitors is currently the only available method of AD treatment. In most countries, AChE inhibitors are the mainstay of symptomatic AD treatment, but in many countries and for early or late stages of the disease, other drugs are approved, too.
  2. Abstract, line 17: This sentence suggests that it is possible to prevent or help prevent AD by taking inhibitors of the acetylcholine esterase. However, clinical trials of acetylcholine esterase inhibitors failed to prevent or delay AD. These drugs have been found effective only for the symptomatic treatment of AD.
  3. Introduction, line 44: It is not quite correct that there is no effective treatment against AD. Some drugs, such as the AChE inhibitors, are moderately effective in improving cognition and activities of daily living in patients with AD, but there are no drugs to cure AD.
  4. Introduction, line 64: The wording "which show AD-preventive properties" implies that preventive efficacy has been proven. But this is not the case. Findings from epidemiological and mechanistic studies suggest that these substances may decrease the risk for dementia and have a role in preventing or delaying dementia, but such efficacy has not been proven yet by clinical trials.
  5. Materials and Methods, lines 108-111: The last two sentences belong to the next paragraph (2.4. Statistical analysis). Some of the information is there already.
  6. Results, Figure 1: It should be indicated for the columns, which pattern represents which brewing time.
  7. Discussion, line 144: IC50 levels of 0.03 mg/ml and 0.06 mg/ml are mentioned here. To what are these quantities related? Milligrams of tea leaves or of certain compounds?

Minor Issues:

  1. Introduction, line 31: Reference No. 1 does not seem to be the appropriate reference for this specific statement.
  2. Introduction, lines 34-38: For the statements related to the role of acetylcholine esterase and butyrylcholine esterase in AD and AD treatment, some seminal papers from the AD literature should be cited, rather than papers from the plant literature which refer to other papers.
  3. Introduction, line 47: Reference No. 13 does not seem to be the appropriate reference for this statement.
  4. Introduction, line 64: Please provide full term "amyloid precursor protein" before using the abbreviation "APP".
  5. Discussion, line 136: Most of the studies carried out up to now were epidemiological studies and mechanistic studies. Only a few clinical trials have been done. The cited review by Polito et al. (2018) does not cover any clinical studies and the data reported by Shen et al. 2015 were from a cross-sectional study. Starting the sentence with "Clinical studies …" puts too much emphasis on "clinical".
  6. Discussion, line 183: Abbreviation for Alzheimer's Disease in capital letters: "AD".
  7. I think, I understand how to read Table S1, but I am not really sure. It is difficult to read, and it takes time to find out how to read it. It would be nice seeing this information presented in a different way that is easier to understand.

Author Response

 Reviewer #1: Manuscript Number: nutrients-730476

General Comments:

The paper reports on the acetylcholine esterase (AChE) inhibitory potential of various types of tea, which was found to be independent from brewing temperature and time. Considering that, in general, the types and quantities of molecules extracted from plant material strongly depend on the extraction conditions, whereas, on the other hand, the brewing of tea is usually done in a non-standardized way, these findings seem important with regard to making use of the health effects of tea.

It is important to strictly distinguish throughout the paper between prevention of Alzheimer's disease (AD) and the symptomatic treatment of AD. Polyphenols and a few other types of molecules, mainly due to anti-oxidant activity or interference with the amyloid cascade, are supposed to have some preventive or disease-modifying effects. AChE inhibition does not seem to have a role in AD prevention or disease-modification, because the known AChE inhibitors did not delay or prevent dementia in controlled clinical trials. During both the introduction and the discussion, the wording should be clear for every substance or substance group, whether it is supposed to have a role in prevention or in symptomatic treatment of dementia. Currently, AChE inhibition has its role in symptomatic treatment of the dementia stage of AD.

Taking into account that the focus of the experimental part of the paper is on AChE inhibition by various types of tea, both the introduction and the discussion should focus more on AChE inhibition than on other mechanisms. Nevertheless, for putting the findings into context, the other beneficial effects of tea should also be mentioned Full Title: Honey as the potential source of cholinesterase inhibitors in Alzheimer's disease. The work is very preliminary, it has several weaknesses. The authors have evaluated a significant number of samples, but unfortunately the results are not relevant.

Specific Comments:

  1. Abstract, line 15: It is not quite correct that administration of AChE inhibitors is currently the only available method of AD treatment. In most countries, AChE inhibitors are the mainstay of symptomatic AD treatment, but in many countries and for early or late stages of the disease, other drugs are approved, too.

AU: Please note that the title of the work was simplified and shortened. The sentence in introduction was changed according to the Reviewer’s suggestion

  1. Abstract, line 17: This sentence suggests that it is possible to prevent or help prevent AD by taking inhibitors of the acetylcholine esterase. However, clinical trials of acetylcholine esterase inhibitors failed to prevent or delay AD. These drugs have been found effective only for the symptomatic treatment of AD.

AU: The sentence was changed according to the Reviewer’s suggestion

  1. Introduction, line 44: It is not quite correct that there is no effective treatment against AD. Some drugs, such as the AChE inhibitors, are moderately effective in improving cognition and activities of daily living in patients with AD, but there are no drugs to cure AD.

AU:  Corrected according to Reviewer’s comment.

  1. Introduction, line 64: The wording "which show AD-preventive properties" implies that preventive efficacy has been proven. But this is not the case. Findings from epidemiological and mechanistic studies suggest that these substances may decrease the risk for dementia and have a role in preventing or delaying dementia, but such efficacy has not been proven yet by clinical trials.

AU: Corrected according to Reviewer’s comment

  1. Materials and Methods, lines 108-111: The last two sentences belong to the next paragraph (2.4. Statistical analysis). Some of the information is there already.

AU: It was corrected as suggested. Lines 108-111 was removed from the text.

  1. Results, Figure 1: It should be indicated for the columns, which pattern represents which brewing time.

AU: It was corrected as suggested. We added legend in Figure 1.

  1. Discussion, line 144: IC50 levels of 0.03 mg/ml and 0.06 mg/ml are mentioned here. To what are these quantities related? Milligrams of tea leaves or of certain compounds?

AU: It was corrected as suggested. We wrote, line 139: “For green and black tea…. IC50 levels of 0.03 mg/ml and 0.06 mg/ml, respectively”.

  1. Introduction, line 31: Reference No. 1 does not seem to be the appropriate reference for this specific statement.

AU: It was corrected as suggested.

The reference has been changed. References No. 1 is now: “Prince, M.; Bryce, R.; Ferri, C. Alzheimer’s Disease International World Alzheimer Report 2011. The benefits of early diagnosis and intervention. Alzheimer’s Disease International 2011”.

We changed references No. 2, we added a paper: “Wortmann, A. Dementia: a global health priority – highlights from an ADI and World Health Organization report. Alzheimer’s Res. Ther. 2012, 4, 40”.

  1. Introduction, lines 34-38: For the statements related to the role of acetylcholine esterase and butyrylcholine esterase in AD and AD treatment, some seminal papers from the AD literature should be cited, rather than papers from the plant literature which refer to other papers.

AU: It was corrected as suggested. The reference has been changed (No. 3-10)

  1. Introduction, line 47: Reference No. 13 does not seem to be the appropriate reference for this statement.

AU: It was corrected as suggested. The reference has been changed. We added a paper:

“Nurk, E.; Refsum, H.; Drevon, C.A.; Tell, G.S.; Nygaard, H.A.; Engedal, K.; Smith, A.D. Intake of Flavonoid-Rich Wine, Tea, and Chocolate by Elderly Men and Women Is Associated with Better Cognitive Test Performance. J. Nutr., 2008, 139, 120–127”.

  1. Introduction, line 64: Please provide full term "amyloid precursor protein" before using the abbreviation "APP".

AU: It was corrected as suggested. Line 62

  1. Discussion, line 136: Most of the studies carried out up to now were epidemiological studies and mechanistic studies. Only a few clinical trials have been done. The cited review by Polito et al. (2018) does not cover any clinical studies and the data reported by Shen et al. 2015 were from a cross-sectional study. Starting the sentence with "Clinical studies …" puts too much emphasis on "clinical".

AU: It was corrected as suggested. Line 132

  1. Discussion, line 183: Abbreviation for Alzheimer's Disease in capital letters: "AD".

AU: It was corrected as suggested. First line 138, now Line 134

  1. I think, I understand how to read Table S1, but I am not really sure. It is difficult to read, and it takes time to find out how to read it. It would be nice seeing this information presented in a different way that is easier to understand.

AU: Significant differences was presented in a different way. We added Table S2 in supplementary material.

Let me thank you for your valuable comments concerning my paper.

Reviewer 2 Report

The manuscript entitled “Regardless of the brewing conditions, various types of tea can constitute a potential source of cholinesterase inhibitors in Alzheimer’s Disease” by Baranowska-Wòjcik Ian et al. focuses on the effect of brewing conditions on acetylcholinesterase activity of various types of tea on the basis that a growing literature has demonstrated the beneficial effect of tea on cognitive functions. I have some comments:

-This communication has an incomplete experiment design

-The authors should report the effect of brewing conditions on phytochemical composition and antioxidant activity of various types

-Several editing errors are in the text

-The discussion is not so detailed and well organized. 

Author Response

Reviewer #2:

Comments and Suggestions for Authors

The manuscript entitled “Regardless of the brewing conditions, various types of tea can constitute a potential source of cholinesterase inhibitors in Alzheimer’s Disease” by Baranowska-Wòjcik Ian et al. focuses on the effect of brewing conditions on acetylcholinesterase activity of various types of tea on the basis that a growing literature has demonstrated the beneficial effect of tea on cognitive functions. I have some comments:

-This communication has an incomplete experiment design

-The authors should report the effect of brewing conditions on phytochemical composition and antioxidant activity of various types

AU: Please note that the title of the work was simplified and shortened. As Reviewer can notice, this is a preliminary study in a form of a short report- type paper. We focused only on one aspect: anti-AChE activity because we wanted to screen and compare a huge number of samples of fruit teas with “standard” teas (produced from C. sinensis). We are at the beginning of our long-lasting project, and the composition of particular teas seems to have secondary, minor importance. At the present moment, we aim at a decent comparison of the most important property of tea infusions- anti-cholinesterase activity. Please note that only a minority of works concerning the anticholinesterase activity of plant infusions, tinctures, extracts etc., uses the “false-positive” sample. In the case of our paper we eliminated the samples that exerted the  “false-positive” effect so we presented only a real activity. This can be a decent base for near-future works of our team.

            As for the antioxidant activity: indeed, it is true that polyphenols are usually both cholinesterase inhibitors and also antioxidants. The antioxidant activity of various polyphenols (phenolic acids and flavonoids) was subject of a considerable number of our previous works (please see scientific databases for publications of D. Szwajgier). However, in presented, new paper we were not interested in the antioxidant activity of samples. This activity of in the case of various fruits (present in tested teas) is very well documented, far better than the anticholinesterase activity of these fruits. Simply saying, we would like to publish a really compact short report about anticholinesterase activity in order not to duplicate the results previously reported by other authors. We can also shorten the introduction chapter, if you wish, in order to remove some parts that discuss other activities of teas, not directly related to anticholinesterase activity.

-Several editing errors are in the text

AU: Editing errors were corrected in the text according to the Reviewer’s remarks (marked with red color)

-The discussion is not so detailed and well organized. 

AU: The discussion was moderately improved according to the Reviewer’s remarks.

Let me thank you for your valuable comments concerning my paper.

Round 2

Reviewer 1 Report

After the splitting of the supplementary table into S1 and S2, it is still difficult to understand what exactly the letters mean.

Looking, e.g., in Table S1 at tea sample 4 and a brewing temperature of 85°C, there is "b" for 5 minutes, "a" for 10 minutes, and "b" for 15 minutes. From which other brewing temperatures does the temperature of 85°C differ with respect to AChE inhibitory activities, if there is a "b", and from which other temperatures does it differ, if there is an "a"?

Looking, e.g., in Table S2 at tea sample 3 and a brewing time of 10 minutes, there are the letters "A", "C" and "D" for a temperature of 100°C. From which other brewing times does the time of 10 minutes differ with respect to AChE inhibitory activities, if there is an "A", and from which other times does it differ, if there is a "B" or a "C"?

Author Response

The Table S1 was changed according to the Reviewer’s suggestion

Reviewer 2 Report

The authors revised the manuscript following reviewers comments.

Author Response

The authors revised the manuscript following reviewers comments.